# Reconciling ASPP-p53 binding mode discrepancies through an ensemble binding framework that bridges crystallography and NMR data

**Te Liu**, **Sichao Huang, Qian Zhang, Yu Xia, Manjie Zhang***, **Bin Sun***

Research Center for Pharmacoinformatics, College of Pharmacy, Harbin Medical University, Harbin, China

* zhangmj454@nenu.edu.cn (MZ); binsun@hrbmu.edu.cn (BS)

**Data Availability Statement:** Scripts that are used to analyze the data are avaiable from https://github.com/bsu233/bslab/tree/main/2023-p53ASPP.

## Abstract

ASPP2 and iASPP bind to p53 through their conserved ANK-SH3 domains to respectively promote and inhibit p53-dependent cell apoptosis. While crystallography has indicated that these two proteins employ distinct surfaces of their ANK-SH3 domains to bind to p53, solution NMR data has suggested similar surfaces. In this study, we employed multi-scale molecular dynamics (MD) simulations combined with free energy calculations to reconcile the discrepancy in the binding modes. We demonstrated that the binding mode based solely on a single crystal structure does not enable iASPP's RT loop to engage with p53's C-terminal linker—a verified interaction. Instead, an ensemble of simulated iASPP-p53 complexes facilitates this interaction. We showed that the ensemble-average inter-protein contacting residues and NMR-detected interfacial residues qualitatively overlap on ASPP proteins, and the ensemble-average binding free energies better match experimental $K_D$ values compared to single crystllgarphy-determined binding mode. For iASPP, the sampled ensemble complexes can be grouped into two classes, resembling the binding modes determined by crystallography and solution NMR. We thus propose that crystal packing shifts the equilibrium of binding modes towards the crystallography-determined one. Lastly, we showed that the ensemble binding complexes are sensitive to p53's intrinsically disordered regions (IDRs), attesting to experimental observations that these IDRs contribute to biological functions. Our results provide a dynamic and ensemble perspective for scrutinizing these important cancer-related protein-protein interactions (PPIs).

## Author summary

Solution NMR and crystallography often yield disparate results when determining the biologically relevant binding modes of protein-protein interactions. Through computational modeling, we successfully reconciled the discrepancies in binding modes reported by NMR and crystallography for the tumor suppressor protein p53 and its regulators, ASPP2 and iASPP. We demonstrated that the controversial binding modes determined by NMR and crystallography for iASPP-p53 are, in fact, the two major binding modes captured in

Simulation trajectories are deposited into a Zenodo repository (https://zenodo.org/records/10349773).

**Funding:** This work was supported by the Harbin Medical University high-level introduction of talent research start-up fund (No.310212000109 to B.S., No.31011210004 to M.Z), the Provincial Basic Research Fund for Universities (No. 2023-KYYWF-0265 to B.S) and the National Science Foundation (No. 22002096 to M.Z). The funders had no role in study design, data collection and analysis, decision to publish, or preparation of the manuscript.

**Competing interests:** The authors have declared that no competing interests exist.

free protein-protein binding simulations. Therefore, both reported binding modes are plausible, and crystal packing influences the preference for the mode captured in crystallography. Our results underscore the concept that proteins function through dynamic ensembles, and under physiological conditions, protein-protein interactions can have more than one binding mode. Using this ensemble binding framework, we explore the roles of disordered regions of p53 in complex formation, demonstrating how the overlooked IDRs can fine-tune protein-protein interactions.

## Introduction

The tumor suppressor protein p53 is a transcription factor that activates genes involved in apoptosis and cell-cycle arrest upon the detection of oncogenic stress [1]. p53 is a signaling hub that can be regulated by numerous proteins (involved in >1000 protein-protein interactions (PPIs) [2]). The main consequence of p53 is cell apoptosis, but it can also cause cell-cycle arrest, depending on specific p53 regulators [3]. ASPP2 and iASPP proteins are two p53 regulators that promote and inhibit p53-dependent apoptosis, respectively [4]. Given the critical role p53 plays in tumors, mechanical insights into the reversal regulatory effects of ASSP2 and iASPP on p53 are of great importance for potential therapeutic developments. However, mechanical understanding is obscured, in part, due to controversial binding modes that have been reported on these PPIs.

So far, structural characterizations have primarily focused on interactions within the folded portions, specifically between p53's DNA-binding domain (DBD) and the ankyrin repeats and an SH3 domain (ANK-SH3) of iASPP/ASPP2 [5, 6] (Fig 1A). The ANK-SH3 domains of ASPP2 and iASPP are highly conserved, sharing approximately 70% sequence identity and nearly identical crystal structures ($C\alpha$ RMSD $\sim$1.24 Å) [6, 7]. Despite this high degree of conservation, crystallography has revealed that when binding to p53's DBD, ASPP2 and iASPP employ entirely different surfaces on their ANK-SH3 domains: ASPP2 primarily utilizes the SH3 domain to bind p53, while iASPP employs a flat surface formed by the parallel ank-repeat helices to interact with p53 (Fig 1B). This significant difference in binding modes has been reported to contribute to ASPP2 and iASPP's opposing regulatory effects on p53. ASPP2 directly displaces DNA from p53's DBD through a competitive binding mechanism [8, 9], whereas iASPP binds to p53, triggering a molecular switch in the p53 L1 loop, interfering with p53's DNA binding [6]. While the general mechanism by which ASPP proteins alter p53's binding capacity to various genes, thus influencing cellular life-or-death processes [1, 6, 8], has been established, this mechanism is challenged by conflicting reports on the binding modes of iASPP-p53 interactions. Ahn et al.'s solution NMR data suggests that iASPP uses a similar binding surface on its ANK-SH3 domain as ASPP2 to interact with p53's DBD [4] (Fig 1B), in stark contrast to the crystallography-determined iASPP-p53 complex (PDB 6RZ3).

Discrepancies between crystallography and NMR structures are frequently encountered, especially when determining biologically relevant PPIs [10–13]. Both methods have their own advantages and shortcomings. While the structures themselves are accurate, the protein-protein complexes determined by crystallography may not be biologically relevant due to crystal packing effects [10]. This is especially true for weak PPIs, as it has been reported that there is a greater than 50% probability that crystallography-determined complexes are not biologically relevant if the dissociation constant ($K_D$) is greater than 100 $\mu$M [10]. In contrast, solution NMR captures protein motion in an aqueous environment and is therefore more functionally related. However, it suffers from inaccuracies in chemical shift assignments and cannot

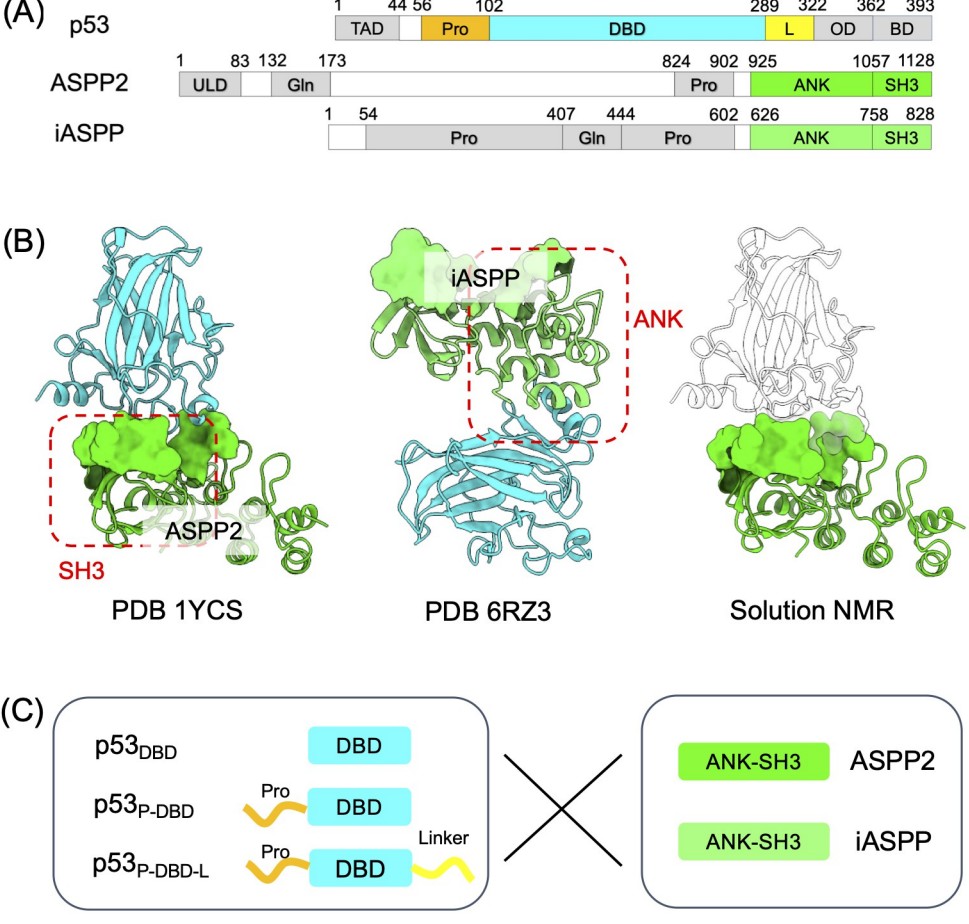

**Fig 1. Structural background of p53-ASPP PPI.** (A) Domain organizations of p53 and ASPP proteins. p53's DBD and ASPP's ANK-SH3 are folded domains whereas the remaining sequences are intrinsically disordered. (B) Crystal structures of ASPP2-p53 complex (PDB 1YCS [5]) and iASPP-p53 complex (PDB 6RZ3 [6]). Solution NMR studies [4] also suggest a iASPP-p53 binding mode that is different from PDB 6RZ3. (C) In this study, we combined all-atom and coarse-grained (CG) MDs plus free energy calculations to explore the binding mechanisms between three p53 constructs (p53$_{DBD}$ p53$_{P-DBD}$ and p53$_{P-DBD-L}$) and the ANK-SH3 domains of ASPP2/iASPP.

directly provide atomistic structures; instead, it serves as constraints to derive structures that satisfy the observed chemical shifts [11]. Solution NMR generally reflects ensemble-average properties, as demonstrated by Pochapsky et al, who used solution NMR to capture multiple p450 protein conformers that are all important to the protein's function [14]. Meanwhile, the binding mode determined by crystallography is more likely to be a subset of the ensemble of binding complexes, as demonstrated by Zuo et al, who used computational methods to show that multiple poses, including the crystallography one, are all consistent with experimental affinities [15]. For our system of interest, whether the observed discrepancies between crystallography and NMR in determining the binding mode for iASPP-p53 are complementary or mutually exclusive is unknown and warrants further exploration.

Reconciling the discrepancy in binding modes can provide valuable insights into the roles of intrinsically disordered regions (IDRs) in regulating the ASPP-p53 PPI. NMR studies have suggested that p53's IDRs play a role in fine-tuning the PPI. Both ASPP and p53 encompass significant portions of their protein sequences predicted as IDRs (Fig 1A and S1 Fig). Ahn

et al. demonstrated that ASPP2 primarily binds to p53's DBD, while iASPP predominantly interacts with a linker region located C-terminal to the DBD domain [4]. This highlights the influence of p53's IDRs in distinguishing between bindings to the highly conserved ANK-SH3 domains of ASPP2 and iASPP. Furthermore, in the case of ASPP2-p53 binding, solution NMR studies conducted by Tidow et al. [8] revealed that, in addition to the interface residues reported in PDB 1YCS, p53 residues outside the interface (such as loop 1 and the C-terminal helix) were implicated in binding ASPP2. Clarifying the relevance of these NMR-based observations to ASPP-p53 functions is expected to be possible if the aforementioned binding mode discrepancy is resolved.

In this study, we performed extensive multi-scale molecular dynamics (MD) simulations to investigate if p53's DBD domain and ASPP2/iASPP ANK-SH3 domain could bind in multiple binding modes, and whether the controversial crystallography- and NMR- determined binding modes can be reconciled under this ensemble binding framework. We also simulated the bindings of different p53 constructs that contain extra IDRs to ASPP, to explore how p53's IDRs fine-tune the ASPP-specific PPIs. We anticipate that our structural characterizations could facilitate thearpeutic developments targeting these important PPIs.

## Materials and methods

### Conventional molecular dynamics (MD) simulations

Simulations were based on the crystal structures of $p53_{DBD}$ in complex with ASPP2 and iASPP's ANK-SH3 domains (PDB IDs are 1YCS [5] and 6RZ3 [6], respectively). The input files for MD were prepared by the TLEAP program from Amber20 package [16]. The system was solvated into a 12Å margin OPC waterbox with 0.15 M KCl ionic strength, and the Amber ff19SB force field [17] was used for protein. The system was first subject to 50000 steps energy minimization with the first 200 steps using the steepest descent algorithm and the remaining steps using the conjugate gradient algorithms. The minimized system was then heated to 300 K via a two-step procedure: 0 to 100 K in NVT ensemble over 0.1 ns followed by 100 to 300 K in NPT ensemble over 0.5 ns. During heating, harmonic constraints with a force constant of 5 kcal/mol/$Å^2$ were introduced onto protein backbone atoms. After heating, a 1 ns equilibrium simulation was performed in the NPT ensemble at 300 K with reduced force constant of 1 kmol/mol/$Å^2$. Three independent 1 $\mu$s long production runs (NPT ensemble, 300 K) were initiated from the equilibrated system. During the simulation, all of the lengths of bonds involving hydrogen were restrained by the SHAKE algorithm [18], and the temperature was controlled by Langevin thermostat [19], and the Particle-Mesh Ewald (PME) method [20] was used to handle long-range electrostatic interactions. The time step was 4 fs after hydrogen mass repartitioning [21] and the nonbonded cutoff was 12.0 Å.

Notably, the bound zinc ion in p53 crystal structures was considered in our all-atom MD simulations. This $Zn^{2+}$ is vital to p53's structure stability and function [22, 23], and our own test (S2 Fig) shows that removing the $Zn^{2+}$ alters the dynamics of $p53_{DBD}$ and changes the conformation of motifs at the PPI interface towards ASPPs. However, since protein force fields generally notoriously handle polar contacts [24], to maintain the coordination of $Zn^{2+}$ to p53's His83-CYS80,142,146 motif, we introduced harmonic constraints between $Zn^{2+}$ ion and the coordinating atoms with a force constant of 300 kcal/mol/$Å^2$ to retain $Zn^{2+}$ binding. The parameters of $Zn^{2+}$, as well as the KCl monovalent ions, were adapted from the Li-Merz dataset [25].

Besides the $p53_{DBD}$, we built two additional p53 constructs that have the IDR regions flanking p53's DBD domain added: The $p53_{P-DBD}$ that has the N-terminal Pro-domain (residues 56–102) added, and $p53_{P-DBD-L}$ that has the additional C terminal linker (residues 289–322)

added. The initial structures of these IDRs were built from sequences using the TLEAP program, and were sampled from 500 ns conventional MD at 300 K in explicit solvent model. While this 500 ns long conventional MD is generally insufficient to thoroughly explore the conformational space of these IDRs, it provides relatively stable structures for the subsequent simulations to start with (S3 Fig). The representative structure of the most-populated cluster was selected, and was then joined to $p53_{DBD}$ using Pymol. Simulations of these p53 constructs' interactions with ASPPs were started from the above mentioned PDB structures and followed the same MD protocol.

## Martini CGMD to simulate the binding process between p53 and ASPP

The Martini3.0 coarse-grained (CG) model [26] was used to simulate the protein-protein binding. Coarse graining of p53 and ASPP all-atom structures into Martini beads were performed using the VERMOUTH program [27], and the DSSP program [28, 29] was used to detect protein secondary structure information. The bound $Zn^{2+}$ ion in p53 was omitted for coarse graining, thanks to the elastic network strategy that can maintain protein tertiary structure. Notably, for p53 constructs, we excluded the IDRs from the elastic network constraints to retain the dynamic nature of the IDRs. We showed in S3 Fig that this strategy indeed allowed p53's IDRs to sample various conformations prior to binding to ASPP2/iASPP. The elastic bond force constants was set as 500 kJ/mol/nm$^2$ and the lower- and upper- elastic bond cutoffs were 5 and 9 Å, respectively. Since long range electrostatic interactions between proteins become centrysymmetric at $\sim 40$ Å [30], the ASPP ANK-SH3 domain was randomly placed >=40 Å away from p53 construct to minimize bias. The system was solvated in a $150 \times 150 \times 150$ Å cubic box with 0.15 M ionic strength using standard Martini water and NaCl models [31]. The system was first energy minimized 2000 steps using the steepest descent algorithm and was then heated to 300 K in the NPT ensemble over 1 ns, with constraints on the protein. The equilibrated system was subject to a 4 $\mu$s production MD in NPT ensemble at 300 K using a 20 fs time step. The temperature control was achieved with the velocity rescale (V-rescale) and pressure was controlled with the Parrinello-Rahman barostat. For each system, 50 replicas of 4 $\mu$s CGMD runs were performed via Gromacs2020.4 [32] to investigate the binding between p53 constructs and ASPP ANK-SH3 domains.

## Umbrella samplings to obtain the dissociation potential of mean force (PMF)

Umbrella samplings were performed to estimate the potential of mean force (PMF) along the dissociation pathway of the p53-ASPP complexes, starting from both crystal complex structures and Martini CGMD sampled complexes. Martini complexes were backmapped into all-atom structures using the BACKWARD.PY [33] script, and were subject to 100 ns explicit solvent MD with $Zn^{2+}$ added to p53 to further optimize the complex following the abovementioned MD protocol. The reaction coordinate (RC) was defined as distance between centers of mass (COM) of p53 and ASPP and was binned into 0.5 Å-wide windows. Samplings in each window was enforced by harmonic constraints with 10 kcal/mol/Å$^2$ force constant. Starting structures used for each window are the last trajectory frames from previous windows. For each window, a 10 ns production MD was performed in NPT ensemble at 300 K. The PMF was estimated by the WHAM program [34], and the errors on the PMF curves were estimated by the built-in Monte Carlo bootstrapping algorithm of WHAM program by setting *num_MC_trials = 100*. Briefly, it generated resamplings of the RC data points and calculated the corresponding PMF profile, and repeated this trials by user-designated times to estimate PMF errors. All MD simulations performed for this work are summarized in S1 Table.

## Sketch-map projections to obtain 2D mode

The sketch-map dimensionality reduction method [35, 36] was employed to project the Martini CGMD sampled ASPP-p53 complexes onto 2D plane. Each trajectory frame was encoded as a high-dimensional space vector, $\vec{X}_D = (D_1, D_2, D_N)$, where $D_1, D_2 \ldots D_N$ are the minimum distances between the backbone beads (Martini BB type) of p53 and ASPP. Beads were chosen from the secondary structure elements (helices and sheets) from p53 and ASPP because inter-protein distances between these rigid structures have low fluctuations compared with loops, and can thus accurately encode the binding configurations [37]. We set N as 21: $D_1$ to $D_{10}$ represents the minimum distance of the 10 residues of p53 to 11 residues of ASPP. Namely, for residue $i$, we first calculated its distance to the selected ASPP residues and assign the smallest value to $D_i$. And $D_{11}$ to $D_{21}$ are the reverse minimum distances, namely, the minimum distance between ASPP residues to p53. Projection follows the principle that points close in the high-dimensional space should be close in the 2D space [35], by minimizing the following stress function [35]:

$$\chi^2 = \frac{\sum_{j \neq i} w_i w_j [F(R_{ij}) - f(r_{ij})]}{\sum_{j \neq i} w_i w_j} \tag{1}$$

where $w_i$, $w_j$ are weights of points, $R_{ij}$, $r_{ij}$ are distances between points in the high- and low-dimensional spaces, respectively. $F$, $f$ are sigmoid functions that map distances between 0 and 1. In our calculations, points weight and sigmoid functions were taken as the default values from the sketch-map tutorial. Notably, such sigmoid functions bias sketch-map's preference for reconstructing connectivities among closer data pairs, while the algorithm's ability of recapturing far-away data pairs in the low-dimension space fades [35, 36]. This may overdampen the presentation of less-similar binding modes after projection.

## Analyses

Analyses were conducted using the CPPTRAJ program [38], MDTraj [39], and Matplotlib libraries. RMSD, RMSF and distance calculations were conducted using CPPTRAJ. 3D space distributions of p53 around ASPP were estimated using the *grid* command from CPPTRAJ. We selected a 0.5 Å spacing to build 3D grids centered on the average COM position of ANK-SH3 domain after aligning the MD trajectories on this domain, and then binned p53 densities into grids, and normalized to standard water density 1.0 g/cm$^3$. Projections of MD trajectories onto 2D plane were done using the MDTraj library which aligns trajectories and extracts the Cartesian coordinate of protein atoms. Inter-protein contact data was calculated using the *nativecontacts* command from CPPTRAJ with default distance cutoff 7 Å. The encounter time was defined as the time when the first inter-protein contact forms during the Martini CGMD. The means and standard error of means of these quantities were estimated via a bootstrapping procedure to obtain 10000 resamplings (sample size equals 50) from the original 50 simulation replicas. Structure were rendered using UCSF ChimeraX [40, 41], and VMD [42]. Scripts supporting this work are available at https://github.com/bsu233/bslab/tree/main/2023-p53ASPP.

## Results

### MD simulations starting from crystallography binding modes fail to capture the experiment-verified inter-protein interactions

Both NMR studies [4] and peptide screening assays [43] have confirmed that p53 utilizes its intrinsically disordered linker (located C-terminal to the DBD domain) to bind to iASPP's RT loop. A designed peptide mimicking p53's linker has been shown to competitively displace p53

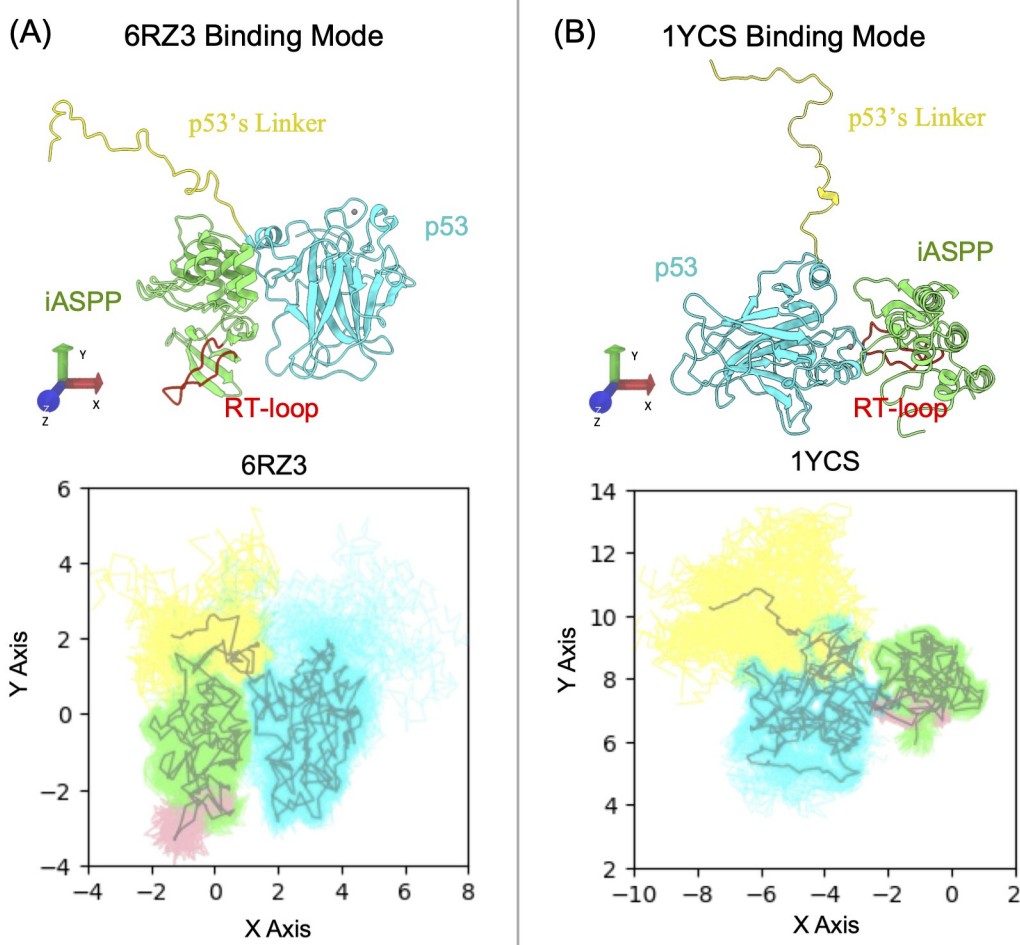

**Fig 2. Examining the interactions between p53's C-terminal linker and iASPP's RT loop.** (A) Samplings of p53's C-terminal linker (yellow) and iASPP's RT loop (red) from $3 \times 1\ \mu$s all-atom MDs starting from the PDB 6RZ3 binding mode. (B) Starting from PDB 1YCS binding mode. For each case, MD trajectories were concatenated and were aligned on p53's DBD against the shown respective reference structure. The X and Y Cartesian coordinates of the protein backbone atoms were used for the projection. ASPP and p53 are colored green and cyan, respectively. The gray lines depict the average structures of protein backbone.

from binding to iASPP, thereby blocking iASPP's regulation of p53 in vitro [43]. We conducted MD simulations to investigate whether p53's linker can interact with iASPP's RT loop in the crystal structure-resolved binding modes, aiming to justify the relevance of these modes as the biological unit. We initiated our simulations from both PDB 1YCS and 6RZ3, assuming that p53 can bind to iASPP via these two modes. The samplings presented in Fig 2 clearly demonstrate that, starting from both crystal structure-resolved binding modes, p53's linker is unable to reach iASPP's RT loop. The discrepancies observed between the MD samplings and experimental findings strongly suggest that iASPP and p53 likely engage in alternative binding modes.

## Ensemble-average interprotein contacts agree with solution NMR revealed PPI interfaces

To explore potential alternative binding modes between p53 and ASPP, we conducted unbiased coarse grained MD to simulate the protein-protein binding processes. We opted for the

Martini 3.0 CGMD model [26] due to its consideration of side-chain packing, which has been shown to have a critical impact on protein-protein interactions [44]. We first simulated the binding between p53$_{\text{P-DBD}}$ and the ANK-SH3 domains of ASPPs. Solution NMR studies conducted by Ahn et al [4] reported chemical shift changes of ASPP residues induced by p53 binding, allowing us to map out the interfacial residues on the ANK-SH3 domains. We therefore calculated the inter-protein contact frequency based on the cumulative Martini CG trajectories and compared the per-residue contact frequency of ANK-SH3 domain residues with the NMR data, as these two quantities are closely related in annotating PPI surface residues.

As shown in Fig 3A, overall the Martini CG captures the major interfacial residues as detected by NMR. To quantify the overlap of CG and NMR, we calculated the RMSE (root mean square error) as $RMSE = \sqrt{\sum_i (CG_i - NMR_i)^2 / N}$, where $NMR_i$ and $CG_i$ are NMR chemical shift changes and Martini contact frequency (average value after bootstrapping,

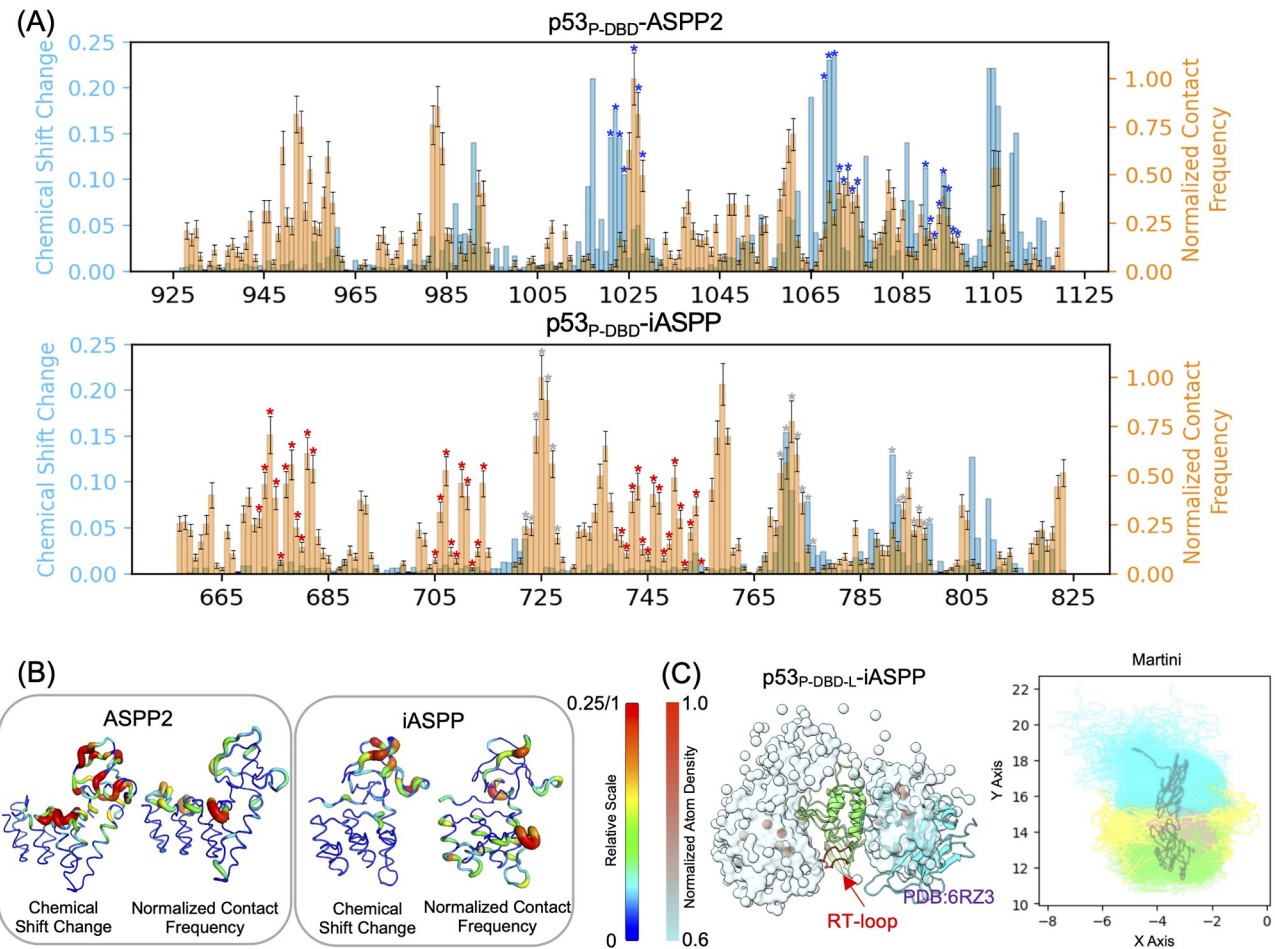

**Fig 3. Martini CGMD simulated p53-ASPP binding complexes.** (A) Comparison of NMR-detected residue chemical shift changes and Martini CGMD sampled residue contact frequency on the ANK-SH3 domains of ASPP. Martini contact frequency was calculated by first summing all contacts involving each ASPP residue and divided by the total number of frames, followed by normalization to the largest value. ASPP residues that are involved in interacting with p53 as determined by crystallography were marked by differently-colored asterisks: blue from PDB 1YCS (ASPP2-p53$_{\text{DBD}}$), red from PDB 6RZ3 (iASPP-p53$_{\text{DBD}}$), and gray from the NMR-suggested 1YCS-like binding mode for iASPP. (B) Mapping the data shown in panel A onto the ANK-SH3 domain structures. (C) Martini CGMD sampled p53$_{\text{P-DBD-L}}$-iASPP complexes. The normalized atoms densities of p53 in the 3D space around ASPP, and the projections of samplings onto the XY plane were shown to the left and right, respectively. The p53's C-terminal linker (yellow) is interacting with iASPP's RT loop (red).

normalized to the highest value observed) for all ASPP residues. We reported RMSE = 0.291 and 0.248 for iASPP and ASPP2, respectively. Our interpretations are CG and NMR data have moderate deviations for both ASPPs, with iASSP being slightly more severe. However, for iASPP, our Martini CG captures almost all of the crystal structure (PDB 6RZ3) determined interfacial residues, as the red "*"-marked residues are encompassed by Martini CG data (Fig 3A). Lastly, although CG and NMR data do show moderate deviations, after mapping them onto ASPP proteins, the overall interfacial regions detected by NMR are qualitatively captured by Martini CG simulations (Fig 3B). In the case of ASPP2, the contact data suggests that the borders between ANK repeats, particularly the ANK3-ANK4 border, exhibit significant contacts with p53. Furthermore, the SH3 domain is largely involved in contacting p53. These contact patterns are consistent with the NMR chemical shift data. For iASPP, both Martini CGMD and NMR data indicate that the SH3 domain is at the protein-protein interface. Additionally, Martini CGMD reports that ANK4 is involved in interacting with p53 (Fig 3B), partially resembling the inter-protein interaction observed in the $p53_{DBD}$-iASPP complex crystal structure (PDB 6RZ3). This suggests that the ensemble of p53-iASPP structures sampled using Martini CGMD may contain the binding mode observed in the crystal structure (which will be discussed in detail later). Overall, we observed that the binding complexes obtained from Martini CGMD qualitatively capture the protein-protein interface reflected in the solution NMR data.

Having validated the Martini CGMD methodology for characterizing PPIs, we proceeded to simulate the binding between $p53_{P-DBD-L}$ and iASPP to investigate whether p53's C-terminal linker interacts with iASPP's RT loop, as experimentally demonstrated [4, 43]. We computed the atom densities of $p53_{P-DBD-L}$ around iASPP, calculated from the accumulated Martini CGMD trajectories, to explore if p53's linker is close to iASPP's RT loop in space. As shown in Fig 3C, overall p53's IDRs form large low-density cloud around iASPP, and particularly encompass the RT loop, with few denser points (normalized density > 0.8) contributing from the folded DBD domain. Projections of these samplings onto a 2D plane further elucidated the specific interactions. These data provide compelling evidence that the simulated binding complexes accurately capture the interactions between p53's linker and iASPP's RT loop, in agreement with experimental findings.

## Ensemble-average binding free energy agrees with experimental affinities

We have demonstrated that the ensemble binding complexes qualitatively recapitulate the results of solution NMR and peptide screening assays. Now, we aim to strengthen these outcomes quantitatively. $p53_{DBD}$ binds to the ANK-SH3 domains of ASPP2 and iASPP with comparable affinities ($K_{D}$s are 26.4 ± 1.7 nM and 23.3 ± 1.6 nM, respectively [7]). Consequently, we calculated the potential of mean force (PMF) for dissociation and compared it against these experimental $K_{D}$s to validate the binding modes.

Based on the dissociation PMF curves (Fig 4A and S4 Fig), we defined the binding free energy as $\Delta G = PMF_{bound} - PMF_{unbound}$, where bound and unbound states were set at $RC = 1.5$ Å and $RC = 14.5$ Å, respectively. With error propagation, we obtained significantly different $\Delta G$ values of −12.73 ± 0.12 kcal/mol and −7.57 ± 0.11 kcal/mol for ASPP2 and iASPP, respectively. This suggested that iASPP exhibits significantly weaker affinity than ASPP2 in binding $p53_{DBD}$. This discrepancy contradicts the comparable experimental $K_{D}$ values. To investigate if starting from multiple binding poses can mitigate this discrepancy, we conducted simulations of the binding process between $p53_{DBD}$ and the ANK-SH3 domains using unbiased Martini CGMD and analyzed the binding modes from various perspectives. We projected the Martini CG trajectories onto a 2D plane using the inter-protein distance and RMSD to iASPP/

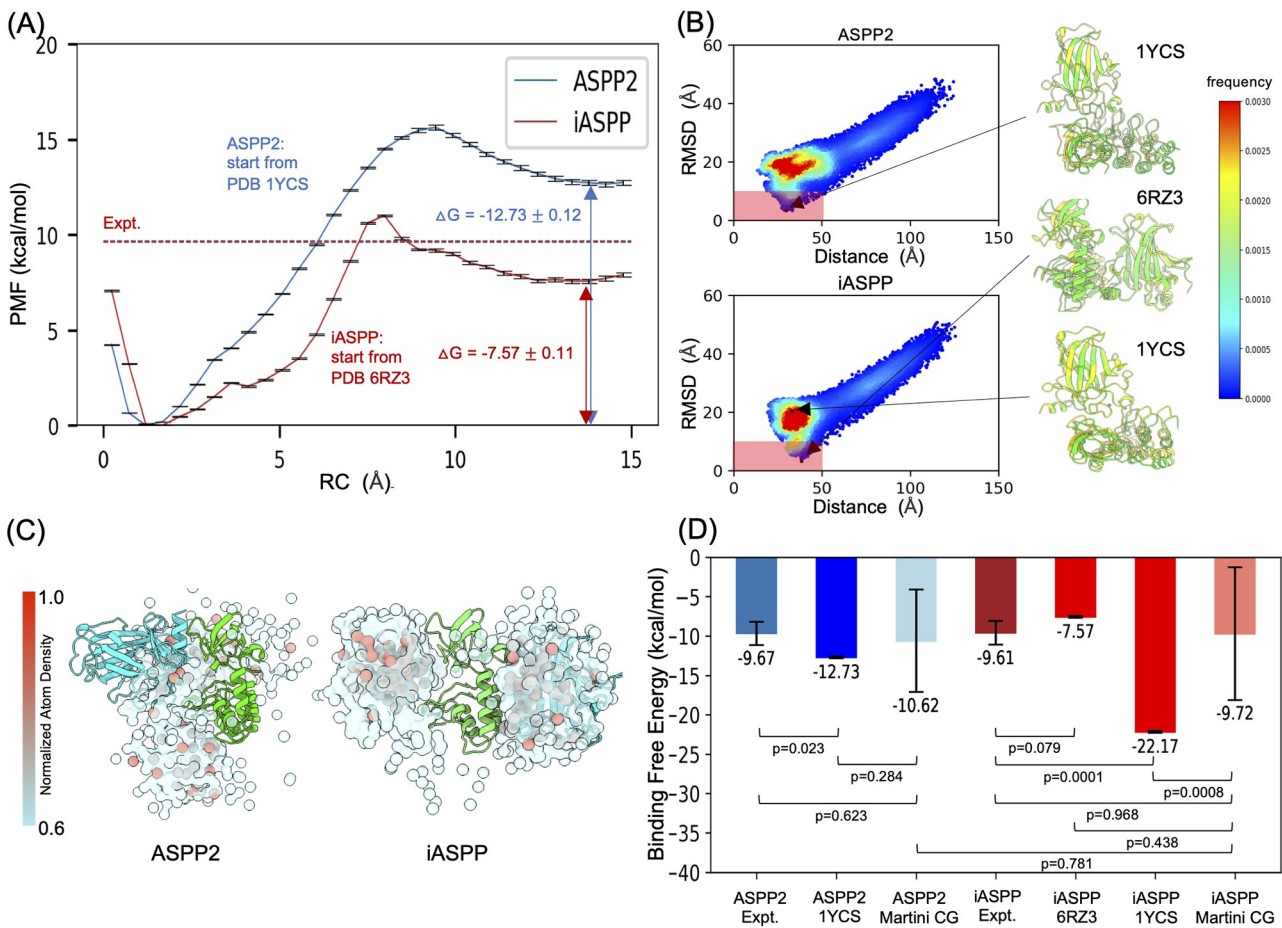

**Fig 4. Energetic characterizations of Martini CGMD simulated binding complexes.** (A) PMF curves of the dissociation process between p53$_{DBD}$ and ASPPs, starting from the crystal-structure binding modes. The PMF errors were estimated by the Monte Carlo bootstrapping procedure of the WHAM program. PMF-derived binding free energy ($\Delta G$) was calculated by defining the bound state at $RC$ = 1.5 Å and the unbound state at $RC$ = 14.5 Å. The experimentally measured $\Delta G$s were shown as red dashed lines. (B) Martini CGMD simulated binding between p53$_{DBD}$ and ASPPs. The trajectories are projected onto the distance (COM of p53$_{DBD}$ to COM of ANK-SH3 domain), and the RMSD (with respect to crystal structures) axes. The color scale means frequency of binding. The red shaded areas highlight the binding poses that are close to crystal structures, and those resemble exactly to crystal structures were superimposed onto crystal structures. (C) Overall atom density distribution of p53$_{DBD}$ (cyan) around ASPP (green) in the 3D space. (D) Comparison of experimentally measured $\Delta G$s, calculated $\Delta G$s based on crystal structures, and $\Delta G$s calculated from multiple representative Martini complexes (detailed PMF curves are shown in S7 Fig). t-tests were performed to assess the statistical significance.

ASPP2-p53$_{DBD}$ co-crystal structures as axes. These two axes are helpful to assess overall distributions of sampled complexes. Interestingly, for both ASPPs, Martini can capture crystal-like binding modes with RMSD as low as 3.2 Å (Fig 4B), although the major simulated bound complexes lie in the 25<distance< 50 Å, and 12 <RMSD< 22 Å regions. To further examine whether the recaptured binding modes are non-specific coincidences resulting from extensive simulation replicas or ASPP-specific binding modes arising from unique protein-protein interactions, we plotted the overall atom density distributions of p53$_{DBD}$ around ASPP (Fig 4C). This clearly shows that p53$_{DBD}$ exhibits different distribution patterns when bound to ASPP2 and iASPP: Martini-sampled p53$_{DBD}$ conformers are continuously distributed around ASPP2, in contrast to the "bi-sided" distributions around iASPP (Fig 4C). This suggests that the sampled binding modes are ASPP isoform-dependent. Notably, the "bi-sided" distributions of p53$_{DBD}$ around iASPP correspond to the major binding modes observed in crystal

structures 6RZ3 and 1YCS, implying that iASPP may bind to p53$_{DBD}$ in two predominant modes.

Lastly, we selected multiple representative Martini CGMD-sampled p53$_{DBD}$-ASPP complexes and performed umbrella samplings to obtain the dissociation PMFs. We employed a rigorous selection procedure to maximize the representativeness of the selected binding complexes (see S5 and S6 Figs). Generally, iASPP has a wider range of affinity binders than ASPP2. Intriguingly, the two iASPP poses based on crystal structures also reflect this wide affinity range: they show a ΔG difference of 14.6 kcal/mol (S4 Fig), resembling the weak and strong binders we selected for iASSP from Martini CGMD-sampled complexes. We performed the t-test between CG simulation-generated poses and crystal structures for iASPP and ASPP2, and t-test between various calculated binding free energies and experimental affinities (Fig 4D). For ASPP2, for which the reported crystallography binding mode is consistent with solution NMR data, the crystal structure based ΔG and CG-simulated poses based ΔG are not significantly different (p = 0.284). For iASPP, while the reported crystal structure (PDB 6RZ3) also does not show significant ΔG difference with that from Martini simulations (p = 0.438), its another NMR-suggested binding mode (PDB 1YCS) is significantly different from the Martini ΔG (p = 0.0008). Importantly, for both ASPP2 and iASPP, ΔGs calculated from crystal structures are significantly different from the experimentally-measured binding affinities. However, the ΔGs based on Martini simulation poses are not statistically different from experimental values, with p = 0.623 and 0.781 for ASPP2, and iASPP, respectively. These data suggest that calculated binding free energies starting from crystal structures do not align with the experimental observations that iASPP and ASPP2 have comparable binding affinities toward p53$_{DBD}$. Instead, the ensemble-average energies based on Martini CGMD-sampled multiple binding poses align with the experimental observations.

## p53's IDRs accelerate the binding rates and further differentiate p53's bindings toward ASPP2 and iASPP

We further investigated the impact of the intrinsically disordered regions (IDRs) in the p53 protein, which constitute approximately 48% of the entire protein (S1 Fig), on the PPIs based on our Martini CG trajectories. We calculated the time of formation of first inter-protein contact to assess how IDRs in our system affect protien-protein binding rates. Comparison between p53$_{DBD}$ and p53$_{P-DBD-L}$ shows that addition of IDRs to p53$_{DBD}$ significantly reduces the mean first passage time for inter-protein contact formation, from approximately 542 ns to approximately 370 ns (Fig 5A). To verify if the greatly shortened "first contact formation time" for IDR-containing constructs may be just due to larger size of the p53 constructs that allow ASPP to more easily interact, rather than is the dynamic nature of IDRs that accelerates the binding, we recalculated the first contact formation time just between p53 DBD domain and ASPP even if p53 IDRs are present. We observed that counting only p53's DBD domain interacting with ASPP still shows comparable first contact formation time as counting the whole p53 constructs. This indicates an enhancement in the association rates facilitated by the presence of IDRs. Moreover, we performed one additional set of Martini CGMD simulations in which p53 IDRs were rigidified via constraints. This was to validate if the dynamic nature of the IDRs affect the binding rates. We observed that treating IDRs as rigid motifs reverse the binding accelerating phenomena, resulting in a much slower binding for the p53$_{P-DBD-L}$. These data suggest that at local structural level, the p53's DBD, regardless the presence of IDRs, is the predominant domain to first interact with ASPPs. More importantly, the shortened first contact formation time, thereby the accelerated binding kinetics of the p53$_{P-DBD}$ and p53$_{P-DBD-L}$ relative to p53$_{DBD}$ are caused by the dynamics of IDRs.

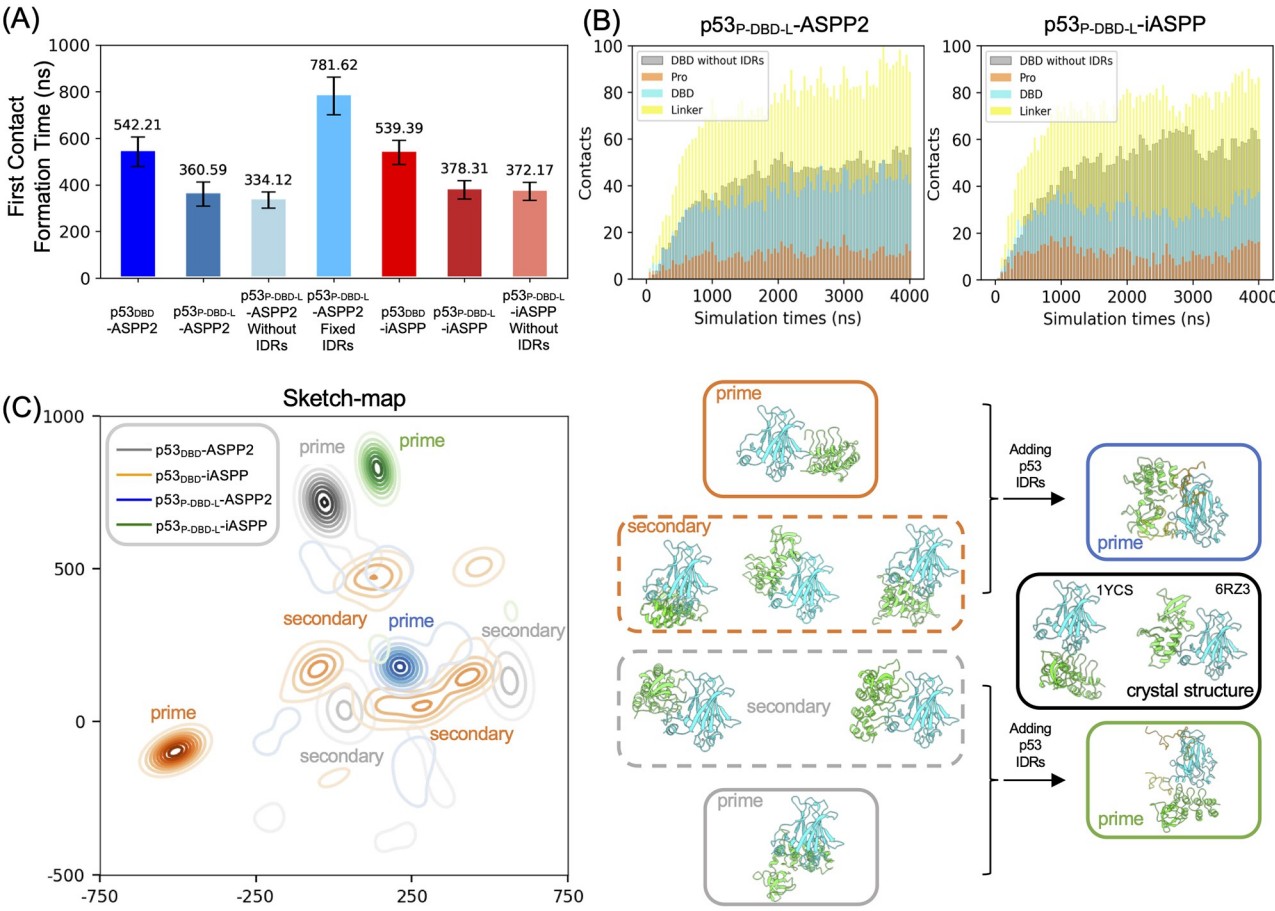

**Fig 5. Effects of p53's IDRs on p53-ASPP binding.** (A) First contact formation times of ASPPs binding to p53$_{DBD}$ and p53$_{P-DBD-L}$. For p53$_{P-DBD-L}$, besides the normal definition of first contact as "any ASPP residue to any p53 residue", we also counted the first contact defined as "any ASPP residue to residues only from p53 DBD domain" (hatched bars). For ASPP2-p53$_{P-DBD-L}$ binding, an additional set of $50 \times 4\,\mu$s Martini CGMD was performed in which p53's IDRs were rigidified by constraints (green bar, using the normal definition of first contact). (B) Average time-dependent number of contacts between ASPP and different p53 domains of p53$_{P-DBD-L}$ calculated from $50 \times 4\,\mu$s Martini CGMD simulations. The contacts based on p53$_{DBD}$-ASPP simulations were also shown as gray bars. (C) Projections of Martini CGMD sampled ASPP-p53 complexes onto 2D plane using the sketch-map dimentionality reduction method. We defined "prime" and "secondary" binding modes to refer to the most highly-sampled states, and the less sampled but still have considerable population states (if present), respectively. Representative complexes were shown right to the sketch-map projection.

Interestingly, while including p53 IDRs greatly accelerates the protein binding kinetics compared to the no IDR cases, the extents of accelerations are comparable for both ASPP2 and iASPP. Thus IDRs do not differentiate p53's binding patterns towards ASPP2 and iASPP kinetically. However, p53 IDRs do influence the specific inter-protein contacts in the simulated complexes. We calculated the time-dependent and p53-domain-specific inter-protein contacts (Fig 5B). Firstly, it shows that p53 DBD's interaction with both ASPPs are attenuated when p53 IDRs are present. Secondly, the degrees of DBD domain de-involvement are different for ASPP2 and iASPP: When binding to ASPP2, p53's DBD domain maintains considerable contacts, whereas for binding to iASPP, the interaction relies less on the DBD domain and is primarily mediated by p53's linker (Fig 5B). These results clearly demonstrate that the presence of p53's IDRs leads to distinct interactions.

Furthermore, the presence of p53's linker serves to differentiate the binding patterns between iASPP and ASPP2 when interacting with p53. To illustrate this, we employed the

sketch-map dimensionality reduction method to project the simulated ASPP-p53 complexes onto a 2D plane. The choice of sketch-map enables precise projection of the relative configurations of ASPP-p53 complexes in 3D space onto the 2D plane. Consequently, points that are close in the 2D plane represent similar ASPP-p53 binding configurations, allowing for straightforward comparison of the simulated ASPP-p53 complexes. As shown in Fig 5C, in the absence of IDRs, the binding of $p53_{DBD}$ to iASPP2 and ASPP2 exhibits moderate overlap, despite differing major binding poses (i.e., high-density areas differ for iASPP2 and ASPP2). Notably, the addition of IDRs to $p53_{DBD}$ leads to a dramatic change in the distribution patterns for both iASPP and ASPP2 on the 2D projection plane. This results in more concentrated and distinct binding patterns for these two ASPPs. To gain structural insights into how the p53's IDRs fine-tune the interactions, we reported representative complexes that correspond to the high-density areas in sketch-map plots (Fig 5C). Consistent with our interpretations on sketch-map, we show that, in the absence of IDRs, although the prime binding modes of ASPP2 and iASPP towards $p53_{DBD}$ are different, their secondary binding modes have similar complexes. Adding IDRs diminishes the overlap and leaves the utterly unlike prime binding modes of ASPP2 and iASPP. These projection data strongly suggest that the inclusion of p53's IDRs further differentiates p53's binding interactions with ASPP2 and iASPP.

## Discussion

Two lines of evidences suggested that the iASPP-p53 binding mode may be different from that resolved in PDB 6RZ3, and this alternative mode is highly similar to the ASPP2-p53 binding mode resolved in PDB 1YCS. Firstly, the isolated iASPP ANK-SH3 crystal structure (PDB 2VGE [7]) was found to use a surface patch, which resembles the ASPP2's binding patch towards p53, to interact with neighbouring unit. Secondly, solution NMR data shows that the p53-induced residue chemical shift perturbations were mapped onto the similar surfaces for ASPP2 and iASPP ANK-SH3 domains [4]. The ongoing debate over the binding mode presents challenges to achieving a comprehensive understanding of ASPP-p53 regulations.

Discrepancies between NMR and crystallography-determined modes are not uncommon in the literature, not only for individual proteins but also for protein-protein interactions (PPIs). While NMR and crystallography-derived protein structures align closely in most cases [11], certain instances can exhibit substantial structural deviations, with backbone RMSD reaching up to 6 Å [12]. This divergence can be attributed to the distinct environments in which the structures are determined, namely, aqueous solution versus crystal lattice. Regarding PPIs, assigning solution NMR-determined PPIs as biologically relevant protein binding modes is generally less contentious. However, the same cannot be said for crystallography-determined PPIs due to the influence of crystal packing. Crystal packing effects can introduce free energy advantages that compete with native PPI formations [10, 45, 46]. Consequently, the relevance of crystallography-determined PPIs to biological functional complexes is affinity-dependent [45]. Strong PPIs (e.g., $K_D < 100$ $\mu$M [45]) are more likely to retain their native binding modes within crystal lattices. In our specific case, the binding affinities of p53 constructs, especially for the $p53_{P-DBD}$, towards ASPP's ANK-SH3 domains can approach the 100 $\mu$M threshold [4, 8]. This brings uncertainty in assuring the crystal structure captured ASPP-p53 complexes as biological-relevant assembly, especially for the iASPP-p53 PPI.

Our simulated p53-ASPP ensemble binding complexes unify the crystallography- and NMR- modes. Firstly, we demonstrated that the ensemble of binding complexes better aligns with experimental observations, supported by two key arguments: 1) the ensemble binding modes satisfy the experimental-confirmed binding patterns while the single crystal structure does not. 2) Calculated binding free energies, using representative complexes from the

ensemble, closely match the experimental $K_D$ values. More importantly, the simulated complexes contain the exact crystal-structure binding modes, and for iASPP, the simulated complexes exhibit "bi-sided" distributions of p53 around the ANK-SH3 domain, which can be treated as two generalized binding modes representing the NMR-suggested and crystallography-captured binding mode, respectively. Therefore, $p53_{DBD}$ binds to iASSP's ANK-SH3 domain in two configurations in solution, and crystal packing shifts the binding equilibrium to the one captured in PDB 6RZ3. This potentially elucidates why iASPP utilizes similar binding modes as ASPP2 towards p53, as suggested by solution NMR, while exhibiting a different binding mode in the crystal structure. As a result, our simulated ASPP-p53 complexes are complementary to the NMR and crystallography characterizations of the p53-ASPP complexes.

IDRs are not merely flexible but have envolved to exert unique functions such as allowing rapid and reversible PPI and acting as recognition motifs [47, 48]. Our simulation data suggest that p53's IDRs play multiple roles in fine-tuning the PPI. IDPs are well-known to accelerate protein binding rates through the "fly-casting" mechanism [49] which states that dynamic IDPs increase the search radius to capture their binding partners. This was reflected in our simulations as we show that adding the N-terminal Pro-domain and C-terminal linker to $p53_{DBD}$ greatly accelerates the binding to both ASPP and iASSP. We additionally confirmed that the rate-accelerating effect originates from the dynamic nature of p53 IDRs by showing that rigidifying the IDRs through constraints totally reverse the rate-accelerating effects. While the rate-accelerating effect brought by p53's IDRs are similar between ASPP2 and iASPP, the simulated protein-protein binding complexes are differently impacted by p53's IDRs. To visualize the difference, we elected to use the sketch-map dimensionality reduction method to project the ASPP-p53 complex conformations onto the 2D plane. We show that adding IDRs to $p53_{DBD}$ significantly reduced the overlap between ASPP2 and iASPP, indicating a difference in the binding interactions sampled.

One angle to explain the functional consequence of differently-sampled ASPP-p53 complexes is through the "refolding of encounter complexes" subprocess of protein-protein binding [44, 50, 51]. Although an objective and uniform structural definition of an encounter complex is lacking, one characteristic of encounter complexes is that they undergo structural refolding to eventually form fully bound (native) states [51]. Therefore, in our Martini CGMD simulations, we assessed whether the sampled complexes exhibited or are undergoing structural reorganizations. We used RMSD to measure structural reorganization in the complexes relative to a particular state, specifically the complex at the initial point of contact. An increasing and continuously changing RMSD indicates ongoing structural refolding, while a plateaued RMSD suggests that the refolding has completed. Conversely, a consistently small RMSD suggests no appreciable refolding is occurring. Interestingly, although we implicitly treated the Martini CGMD sampled complexes as "fully bound complexes" in previous sections, Fig 6 suggests that these simulated complexes were experiencing different degrees of refolding. While in some cases the ASPP and p53 did not refold (e.g. bind and stick together), in most simulation trajectories, these two protein had, and would likely to continue, the refoldings. We speculate that over much longer time-scales, most of our Martini CGMD sampled complexes would continue to refold and converge to fewer binding modes that are more biologically-relevant. From this perspective, the Martini CGMD complexes can be loosely treated as "encounter complex" that would continue their journey to the native modes. In this context, our Martini CGMD amounts to complete the diffusion of protein partners to form the encounter complexes [44], which await the rearrangements to reach native PPI state, such as rolling on another protein's surface [52]. Since we have proved that p53's IDRs can fine-tune the protein-interaction patterns, then it is likely that adding IDRs can change the portions of

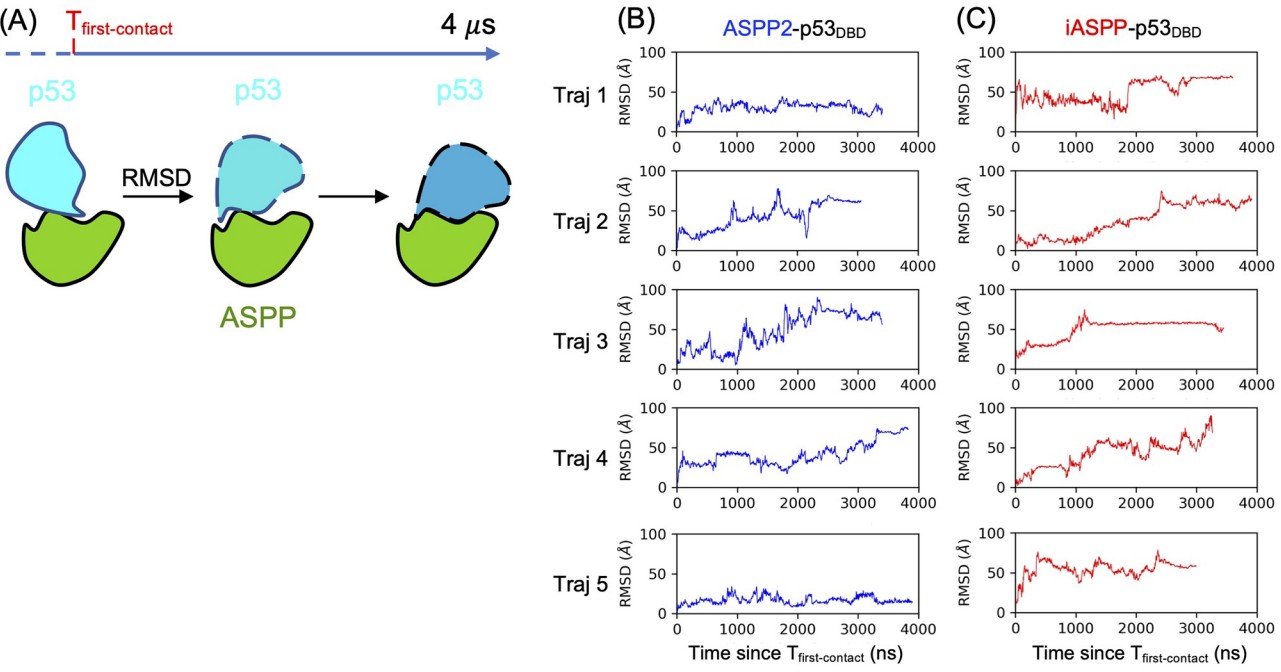

**Fig 6. Assessing the refolding of Martini CGMD sampled ASPP-p53 complexes.** (A) Refolding is gauged by a RMSD metric, namely, starting from the moment when ASPP and p53 first contact, the subsequent complexes' RMSDs with respect to the first contact complex were calculated. (B-C) Five Martini CGMD trajectories were randomly selected to assess the refolding for ASPP2 and iASPP in the binding of $p53_{DBD}$, respectively.

encounter complexes that can successfully reach the destined states. This is because the refolding process is conditional, and occurs only on subsets of the encounter complexes that satisfy certain criteria such as shape complementarity [52]. In this regard, adding IDRs to p53 increase the size of the candidate pool from which more complexes can refold to the native PPI state. Additionally, treating the Martini CGMD sampled complexes as encounter complexes can possibly explain the moderate mismatch between Martini and NMR data in Fig 3A, as those simulated complexes need refold to reach the NMR-detected complexes which are generally treated as fully bound complexes.

An interesting observation in our work is that the iASPP crystal binding mode PDB 1YCS exhibits a stronger affinity than both the Martini CG samples and the PDB 6RZ3 mode. This could be possibly explained via a multi-funnel protein-protein binding free energy landscape model. Although using thermodynamic stability to classify binding complexes into encounter complexes and fully bound states is challenging because the energetics stabilizing encounter complexes are typically case-dependent, non-specific, and transient, and are difficult to characterize [54]. Nevertheless, it is widely accepted that fully bound states are thermodynamically more stable than encounter complexes, and the protein-protein binding free energy landscape is funnel-like [51]. By employing a multi-funnel landscape model that represents different biological relevant bound states, the 1YCS mode can be treated as a biologically relevant mode situated at the funnel bottom. In contrast, our Martini samples contain encounter complexes (as shown in Fig 6) that average down the binding affinities. The weaker 6RZ3 binding mode can be speculated as a specific encounter complex to another funnel (rather than 1YCS) but is trapped in the crystal environment.

However, the mechanism by which ensemble binding of PPIs exerts its biological functions remains elusive, and addressing this question requires further structural characterizations

encompassing the entire ASPP and p53 proteins. Our argument is supported by several studies demonstrating that motifs distant from the folded domains contribute to the protein's biological function. For example, peptide screening assays have shown that ASPP2's Pro-domain can auto-inhibit the ANK-SH3 domain, competing with p53's binding [47]. Additionally, an ASPP2 iso-form with the N-terminal region deleted inhibits p53-dependent apoptosis, exhibiting a completely different regulatory effect on p53 compared to full-length ASPP2 [55]. Even for ASPP1 and ASPP2 proteins, both of which promote p53-dependent apoptosis and have nearly identical sequences and folds in their ANK-SH3 domains, there is a $\sim$10-fold difference in their affinities for p53, with $K_D = 5$ $\mu$M for ASPP2 and $K_D = 0.5$ $\mu$M for ASPP1, respectively. However, ASPP2 can displace the DNA promoter from the p53-DNA promoter complex, while ASPP1 cannot [9]. This contradiction, where ASPP2 has weaker affinity with p53 but can still displace the DNA promoter, implies that other parts of ASPP are involved in *in vivo* regulation. There-fore, a comprehensive understanding of the contrasting regulatory effects of ASPP2 and iASPP on p53 requires structural and biological investigations at the whole protein level in PPIs.

## Conclusion

In this study, we performed multi-scale MDs and free energy calculations to characterize the PPI of two proteins that of great therapeutic potentials, the p53 and its regulator(s), iASPP/ ASPP2. Our work focus on reconciling the discrepancy between NMR- and crystallography-determined binding modes for the iASPP-p53 PPI. We used unbiased Martini3.0 CGMD to simulated the protein-protein PPI, and achieved the ASPP2 and iASPP-specific ensemble binding modes that can unify the reported binding mode discrepancy. We first validated the ensemble binding modes by showing that the ensemble-average inter-protein contacts are well aligned with solution NMR-detected chemical shift perturbations caused by p53. Furthermore, the ensemble-average PPI binding free energies agree with experimental $K_D$s. Notably, we observed that the crystallography-determined binding modes were recapitulated within the ensemble-binding modes, indicating that the crystal packing environment may influence the equilibrium of binding modes toward the configuration captured by crystallography. Although linking these ensemble binding modes to actual biological functions poses a challenge, we pro-vided evidence that the ensemble binding modes are highly sensitive to p53's intrinsically dis-ordered regions (IDRs). The inclusion of IDRs accentuated the differences in binding modes exhibited by p53 towards ASPP2 and iASPP. This sensitivity underscores the biological signifi-cance of ensemble bindings and is consistent with observations indicating that the entire pro-teins, rather than just their folded domains, contribute to biological functions.

## Supporting information

**S1 Table. Molecular dynamics simulations performed in this study.**
(PDF)

**S1 Fig. The intrinsically disordered domains of p53 as predicted by the online-website PONDR [http://pondr.com].** The Pro-domain and linker-domain flanking $p53_{DBD}$ are disor-dered.
(TIF)

**S2 Fig. Effects of $Zn^{2+}$ on simulation results.** (A) Superimposing MD-sampled $p53_{DBD}$ con-formations, w/ $Zn^{2+}$ (blue) and w/o $Zn^{2+}$ (red), on crystal structure (green). (B) RMSD and RMSF of $p53_{DBD}$ from MD simulations w/ and w/o $Zn^{2+}$. (C-D) PMF of $p53_{DBD}$ disassociation from ASPP2 and iASPP w/ and w/o $Zn^{2+}$. Errors bars were drawn in black, and were estimated by the built-in bootstrap error analysis of the WHAM program (*num_MC_trials* = 100). The

relative small errors ($\sim 0.1$ kcal/mol) reflect that our 10 ns samplings per window in the umbrella sampling is sufficient.
(TIF)

**S3 Fig. Samplings of p53 IDRs in isolated state and when attached to the DBD domain.** (A) RMSD and RoG (radius of gyration) of isolated p53 IDRs (Pro-domain and linker-domain) in 500 ns conventional all-atom MD. (B) Structures extracted from the last 10 ns trajectory of 500 ns MD at every 2 ns. (C) Schematic diagram showing the elastic networks that constrain only the DBD domain of p53$_{P\text{-}DBD\text{-}L}$ in the Martini CG simulations. (D) Selected Martini CGMD trajectories showing the highly dynamic IDRs of p53 prior to binding ASPP. During the 4 $\mu$s long Martini CG simulations, p53's IDRs sampled various conformations before binding ASPP.
(TIF)

**S4 Fig. PMFs of iASPP-p53 dissociation starting from the crystallography-determined binding mode (PDB 6RZ3), and from the PDB 1YCS binding mode (assuming iASPP and p53 can bind in this mode).**
(TIF)

**S5 Fig. 20 representative p53$_{DBD}$-ASPP2 complexes sampled by Martini CGMD.** The pairwise RMSDs of ASPP2 (after aligning on p53) were shown on the right to illustrate the structural difference of ASPP2. Procedure for selecting the complexes is: A center of mass (COM) distance criteria ($<30$ Å) and RMSD criteria ($<50$ Å, with respect to PDB 1YCS) were applied on the accumulated Martini trajectories to filter out frames that do not have p53 and ASPP contacted. The surviving trajectory frames were then aligned on p53$_{DBD}$ and were subject to following processes: ASPP protein densities around p53$_{DBD}$ were calculated using the *grid* command from the CPPTRAJ program. Regions have high relative density ($> 0.6$) were identified as the most probable binding patterns. The COM of ASPP was drawn around p53 for each frame. For those frames their COMs are located within the high density regions, they are identified as candidates. 20 frames were randomly picked out from the candidate pool.
(TIF)

**S6 Fig. 20 representative p53$_{DBD}$-iASPP complexes sampled by Martini CGMD.** The pairwise RMSDs of iASPP (after aligning on p53) were shown on the right to illustrate the structural difference of iASPP.
(TIF)

**S7 Fig. PMF curves of representative Martini CGMD sampled complexes.** (A) Umbrella sampling using COM-COM distance as CV can lead to protein gliding on another protein's surface instead of driving complex directly disassociate. We show that during the umbrella sampling, a successful disassociation has inter-protein contacts smoothly disappear as window number increases, indicating a clean one-way disassociation, while for a "gliding" event, newly formed inter-protein contacts are keep emerging as window number increases. (B-C) For the 20 representative Martini CGMD-sampled complexes for each ASPP protein, the PMF curves for successful dissociations (n = 13/20 for ASPP2, and n = 11/20 for iASPP), and for "glidings" are given.
(TIF)

## Author Contributions

**Conceptualization:** Manjie Zhang, Bin Sun.

**Data curation:** Te Liu, Sichao Huang.

**Formal analysis:** Te Liu.

**Funding acquisition:** Manjie Zhang, Bin Sun.

**Investigation:** Te Liu, Bin Sun.

**Methodology:** Sichao Huang, Qian Zhang, Yu Xia, Bin Sun.

**Project administration:** Manjie Zhang, Bin Sun.

**Resources:** Qian Zhang.

**Software:** Sichao Huang, Qian Zhang, Yu Xia.

**Supervision:** Manjie Zhang, Bin Sun.

**Validation:** Bin Sun.

**Writing – original draft:** Te Liu, Bin Sun.

**Writing – review & editing:** Manjie Zhang, Bin Sun.

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
