## [Decision Letter · Decision Letter 0]

1 Nov 2023

Dear Prof. Sun,

Thank you very much for submitting your manuscript "Reconciling ASPP-p53 Binding Mode Discrepancies through an Ensemble Binding Framework that Bridges Crystallography and NMR Data" for consideration at PLOS Computational Biology. And thank you very much for your patience in waiting for this decision.

As with all papers reviewed by the journal, your manuscript was reviewed by members of the editorial board and by several independent reviewers. In light of the reviews (below this email), we would like to invite the resubmission of a significantly-revised version that takes into account the reviewers' comments.

The manuscript has been evaluated by two expert reviewers. Both have acknowledged that the study addresses an important scientific question related to the binding of tumor suppressor p53 transcription factor to two interaction partners ASPP2 and iASPP which play opposite roles in the regulation of p53 mediated apoptosis. The relevance of the study is augmented by the existing discrepancy between different sets of experimental data regarding the binding modes analyzed. While one reviewer only raised minor points to be addressed, the other reviewer raised some substantial issues that all need to be carefully addressed before we can reconsider a revised form of the manuscript.

We cannot make any decision about publication until we have seen the revised manuscript and your response to the reviewers' comments. Your revised manuscript is also likely to be sent to reviewers for further evaluation.

Sincerely,

Vlad Cojocaru, Ph.D.

Academic Editor

PLOS Computational Biology

Arne Elofsson

Section Editor

PLOS Computational Biology

Reviewer's Responses to Questions

**Comments to the Authors:**

Reviewer #1: General assessment:

This paper presents a computational study of ASPP2 and iASPP binding to the DBD of p53. The motivation for the study is to reconcile an apparent contradiction in the X-ray crystallography and NMR data about the binding interface, and simulations present an appropriate way to do this. The paper also examines the effect of adding the p53 IDRs on to the DBD construct to see how they affect binding, which is also very interesting since this complex with the IDRs present is likely hard to study experimentally.

While the data presented on the binding modes seem to explain the discrepancy by demonstrating multiple binding modes for iASPP, there are several major shortcomings in how the data is analyzed and presented. This is primarily with respect to clearly demonstrating the significance of the results with respect to a negative control. The results on the effects of the IDRs are less thorough and are hard to interpret. This part of the study feels incomplete. It could be that more simulations are needed to fully understand how the flanking IDRs effect ASPP binding to p53-DBD.

Major comments:

1. For the simulations of the IDRs alone, what method was used to determine that sampling was sufficient? 500 ns does not seem long enough to capture the structural ensemble of a 48-residue disordered sequence without employing an enhanced sampling method.

2. Figure captions in general should include more information about what is being shown in the figures.

3. Figure 3a&b: While the results somewhat overlap for NMR and CG simulations, it is not clear how good the agreement is, since there are some bars which don’t overlap and it is hard to see where the differences are. It would be helpful to use a quantitative metric to capture the overall agreement and have a control such as comparing the NMR data to the crystal structure to show that these interfaces agree better with the NMR data than the crystal structure interface. Additionlly, error bars could be constructed to quantify the variation between CG simulations to give a sense of the precision of the simulation results.

4. Figure 4b: It is surprising that the binding plots for ASPP2 and iASPP look so similar especially since the RMSD on the y-axis is with respect to a different crystal structure in each case. This does not seem consistent with the differences between the binding that are shown in Figure 4 c. Perhaps this is related to a distinction between the encounter complex and the fully bound state, but this discrepancy should be explained. Additionally, the caption should mention whether it is free energy or frequency that is being plotted and what the color scale means.

5. Figure 4d: It would be helpful to include the delta G values from the PMFs starting from the crystal structures for comparison. The caption also needs to be expanded to explain what the black dashed lines are and the line labeled “ns”.

6. Page 9: The number of weaker binders for ASPP2 and iASPP in Figure 4d is actually about the same. ASPP2 seems to have more tighter binders. A t-test to see if there is a statistically significant difference between the binding free energies for the crystal structures and those from the CG simulation-generated poses would help show that the simulation poses are actually more consistent with experimental binding data than crystal structures.

7. Additionally, it would be interesting to know if there was a free energy difference for the two different iASPP poses, which are presumably included in the different binding complexes sampled in Figure 4d.

8. Figure 5a: The encounter time measurement appears to be based on any contact between p53 and ASPP, but this leads to inconsistency because the simulations with the p53 IDRs have more residues that p53 could use to interact with the ASPP. To determine whether the IDRs are actually affecting binding kinetics, just the contacts between p53-DBD and ASPP could be used even in the simulations with the IDRs present. This would allow a consistent comparison between the simulations. Additionally error bars for this plot calculated based on variation between the independent simulations would be helpful, or a t-test to show there is a significant difference.

9. Figure 5b: The number of contacts with the DBD without the IDRs there should be shown on the plot for comparison. As it is, it is hard to see that the IDRs change the amount of contact with the DBD for iASPP.

10. Fig 5C: The sketch plot is helpful for seeing that the binding pose changes, but all structural information is obscured by the choice of reaction coordinates. It would be helpful to have an example pose shown for each of the highly sampled states. It would be informative to see how the most common binding poses with the IDRs present compare to the binding poses discussed above in Figure 4b&c.

11. Page 10, paragraph 1: The discussion of binding kinetics is confusing. It is stated that the p53 construct with the IDRs binds more quickly in the CG simulations, but then that the addition of IDRs does not significantly alter binding kinetics. The discrepancy here should be explained.

12. Based on the discussion it seems that the authors are making a distinction between the protein-protein interactions that they observe in their simulations which are fully bound and those that are part of the encounter complex, during the binding process. However, it is not clear how they are distinguishing between these two states and which results are focused on the bound state or the encounter complex in figures 3, 4, and 5.

13. Page 12, paragraph 2: In the discussion it says that the KDs are in the uM range, but earlier it says they are between 20 and 30 nM, not near the 100 uM threshold. This discrepancy should be explained.

14. Page 12, paragraph 4: In the discussion the authors argue that this study demonstrates the fly-casting method because the IDRs accelerate binding. However, it is not clear that the increase in binding rate would be significant at relevant concentrations, and may be more due to the size of the IDRs than the fact that they are disordered. The difference in the interaction contacts due to the addition of the IDRs is more interesting, but is hard to interpret due to the opacity of the sketch-map presentation of the data. Additionally, the authors state that these states represent the encounter complex but it is not clear how they are defining that.

15. Page 12, paragraph 5: The argument around IDRs changing the “effective concentration” of the binding partner is unclear. The authors may be referring to the idea that the protein will be more likely to stay in contact with the binding partner instead of diffusing away because the IDRs will facilitate contact with the binding partner, but this is not sufficiently explained. Where the encounter complex fits in to the data presented is also unclear.

Minor comments

1. Abstract: Unclear what “the ensemble-average inter-protein contacting residues and NMR-detected interfacial residues align well with ASPP proteins” means. Do you mean that these groups of residues align well between the two ASPP proteins?

2. Page 3, paragraph 1: “Consequence of p53 activation is majorly cell apoptosis, but can also cause” should be “The main consequence of p53 is cell apoptosis, but it can also cause”

3. Page 3, paragraph 1: “are much wanted yet are obscured by lacking of definite structural characterizations” should be rephrased.

4. It is not mentioned if the particle-mesh Ewald method is used to handle long-range electrostatic interactions in the all-atom simulations.

5. Figure 3: The layout of the figure is confusing. It was not clear at first that each panel contains a top and bottom part. A border around each panel might help. For panels a & b, the caption should read “Normalized contact frequency” rather than “Normal contact”.

6. Figure 3 caption: Each panel should be described in the caption. The structures in the lower parts of panels a & b are not described. The meaning of each of the colors should be explained, for example, in panel c there is yellow in the lower panel but not in the upper panel so it is difficult to connect what is being shown in each panel.

7. Page 12, paragraph 3, first sentence: There is a typo “Our simulated p53-ASPP ensemble binding complexes unify the and crystallography- and NMR- modes” should be “Our simulated p53-ASPP ensemble binding complexes unify the crystallography and NMR binding modes.”

8. Page 12, paragraph 4, first sentence: “reversal” should be “reversible”.

9. Page 12, paragraph 4: “which states that dynamic IDP increases the searching radius to capture binding partner” should be “which states that dynamic IDPs increase the search radius to capture their binding partners”

10. Page 12, paragraph 4, last sentence: “indicating more different samplings..” should be “indicating a difference in the binding interactions sampled.”

11. Page 12, paragraph 5, last sentence: “that from which” should be “from which”.

Reviewer #2: In their manuscript "Reconciling ASPP-p53 Binding Mode Discrepancies through an Ensemble Binding

Framework that Bridges Crystallography and NMR Data", Liu et al. use a combination of molecular dynamics simulations to investigate the binding mode of two p53 regulators, ASPP2 and iASPP, to the transcription factor p53. The authors find conventional simulations, based on crystallographically obtianed structures, do not result in functional binding of iASPP to p53, and differences in binding free energies of iASPP and ASPP2 that do not agree with experimentally determined binding affinities.

Additional coarse-grained simuations that probe more different binding modes, in constrat, capture the binding modes observed in the Xray structures. A significant differnce between the two regulator proteins is the observation of onny one grooup for ASPP2 but two groups of binding for iASPP, each of the latter two groups representing one of the binding modes observed in the Xray structures.

Binding free energy calculations, launched from an ensemble of bound complexes find binding affinites with a different spread but similar averages, which is in better agreement than those binding free energies determined by simlations launched from single structures.

The authors further more show that intrinsically disordered regions of p53 have a significant impact on the binding modes of the two regulator proteins.

The manuscript is well written and a valuable contribution in several aspects. First, it provides insight into the binding modes of the two regulator proteins on p53. Second it shows an impact of, often overlooked, disordered regions of p53 in complex formation. And third, it nicely demonstrates that, at least in solution/under physiological conditions, there is more than one "right" answer, i.e. binding mode, and the importance of studying ensembles.

There are some points that need to be improved before the manuscript is acceptable for publication:

1) The authors state the respective binding free energies of −7.57 kcal/mol and −12.65 kcal/mol, are significantly different. There are no error estimates provided for the values (nor are any shown in the PMF plots), so it is impossible to tell whether this difference is indeed significant (larger than the estimated error). The PMF plot suggest so, but mentioning the estimated error in the text helps the reader.

2) HOw have error estimes been calculated? That is not given in the methods section.

3) Figure 4(B) needs a colour bar tha explains the meaning of the blue, greenish to red areas.

4) What does ns mean in Figure 4(D)?

5) What are the error estimates for encounter times (Fig. 5A) ) ?

6) The PMFs shown in Fig S2 may have error estimates shown as shaded area, but this is unclear to see and not mentioned in the caption.

**Have the authors made all data and (if applicable) computational code underlying the findings in their manuscript fully available?**

Reviewer #1: **No: **The authors state that they will deposit the input files, scripts, and trajectories, but have not yet.

Reviewer #2: Yes

PLOS authors have the option to publish the peer review history of their article (what does this mean?). If published, this will include your full peer review and any attached files.

Reviewer #1: No

Reviewer #2: No
---

## [Decision Letter · Decision Letter 1]

8 Jan 2024

Dear Prof. Sun,

Thank you very much for submitting your manuscript "Reconciling ASPP-p53 Binding Mode Discrepancies through an Ensemble Binding Framework that Bridges Crystallography and NMR Data" for consideration at PLOS Computational Biology. As with all papers reviewed by the journal, your manuscript was reviewed by members of the editorial board and by several independent reviewers.

The reviewers appreciated the revisions performed and concluded that the the manuscript has improved significantly. While one Reviewer recommends acceptance of the manuscript without any further changes. the other Reviewer raised some minor concerns. We are likely to accept the manuscript provided you address these remaining concerns.

Sincerely,

Vlad Cojocaru, Ph.D.

Academic Editor

PLOS Computational Biology

Arne Elofsson

Section Editor

PLOS Computational Biology

Reviewer's Responses to Questions

**Comments to the Authors:**

Reviewer #1: The authors have made substantial improvements to the manuscript particularly with respect to reporting uncertainty. The paper’s results are very interesting and relevant to the field. I have a few additional comments on the revisions.

1. Page 12, line 523: Though it is arguably a qualitative question, I do not see that the RMSD and radius of gyration of the IDRs are approaching a steady value over the 500 ns conventional MD simulations. It is hard to assess this with only a single simulation performed. Since the goal of these simulations was just to generate a reasonable starting structure for these IDRs, I suggest the authors remove this statement.

2. Figure 3A: The values reported are normalized to the highest value observed, but all the normalized values are below 0.25. I would expect to see a value of 1 as the maximum value observed for both the chemical shift change value and the contact frequency. This should be clarified. Possibly the chemical shift change was not normalized and the contact frequency was normalized to have the same maximum value as the chemical shift change.

3. Figure 3A: The left axis should be labeled “chemical shift change” or “normalized chemical shift change”

4. Figure 3A: Although the NMR data does not perfectly match the Martini simulation data, this could be explained by the idea raised in the discussion that the Martini ensemble represents the encounter complex rather the fully bound state. This could be addressed in the discussion.

5. Page 17, lines 693-698: I suggest the authors rephrase to say that iASPP has a wider range of affinity binders than ASPP2 rather than mentioning a distribution or affinitity spectrum since that is no longer shown in the figure.

6. Page 17: It would be interesting to comment more on the fact that the 1YCS binding pose is a stronger binder than the 6RZ3 binding pose or the Martini CG ensemble for iASPP. Does this indicate that the 1YCS binding pose is more like the fully bound state and the Martini CG binding poses represent the encounter complex.

7. Fig 4D: This figure is much clearer than the previous figure. It would be nice to also include the p-values comparing the experimental data to the calculated delta Gs from the crystal structures.

8. Page 19, lines 739-741: The discussion about kinetics is still a bit confusing in the text. I suggest the authors use wording similar to their response to my comment 11 to explain that the adding IDRs does not cause a difference in kinetics between ASPP2 and iASPP.

9. Figure 5C: It would be nice to include the crystal structures in the sketch-map for reference.

10. Page 21: I’m confused about how there seems to be one primary binding mode for iASPP in figure 5C but in figure 3C there are multiple binding modes

11. Page 22-23: It would be helpful to explain the refolding idea more, including how the authors determined when refolding occurred exactly. I am not sure if it was determined by when the RMSD plateaued.

12. Page 23: On line 843 the authors mention “portions of simulated complexes” and line 844-845 “subsets of the encounter complexes.” It is not clear if these are referring to an ensemble of encounter complexes or literally part of the complex structure.

Reviewer #2: The authors have addressed all my concerns.

Also, with the help of the suggestions by the other reviewer, the manuscript has been improved significantly and is now acceptable for publication..

**Have the authors made all data and (if applicable) computational code underlying the findings in their manuscript fully available?**

Reviewer #1: Yes

Reviewer #2: None

PLOS authors have the option to publish the peer review history of their article (what does this mean?). If published, this will include your full peer review and any attached files.

Reviewer #1: No

Reviewer #2: No

Figure Files:

Data Requirements:

Reproducibility:

References:

---

## [Editor Report · Decision Letter 2]

24 Jan 2024

Dear Prof. Sun,

We are pleased to inform you that your manuscript 'Reconciling ASPP-p53 Binding Mode Discrepancies through an Ensemble Binding Framework that Bridges Crystallography and NMR Data' has been provisionally accepted for publication in PLOS Computational Biology.

Best regards,

Vlad Cojocaru, Ph.D.

Academic Editor

PLOS Computational Biology

Arne Elofsson

Section Editor

PLOS Computational Biology

---

## [Editor Report · Acceptance letter]

2 Feb 2024

PCOMPBIOL-D-23-01481R2 

Reconciling ASPP-p53 Binding Mode Discrepancies through an Ensemble Binding Framework that Bridges Crystallography and NMR Data

Dear Dr Sun,

I am pleased to inform you that your manuscript has been formally accepted for publication in PLOS Computational Biology. Your manuscript is now with our production department and you will be notified of the publication date in due course.

With kind regards,

Anita Estes
